# Luteolin Alleviates Cadmium-Induced Kidney Injury by Inhibiting Oxidative DNA Damage and Repairing Autophagic Flux Blockade in Chickens

**DOI:** 10.3390/antiox13050525

**Published:** 2024-04-26

**Authors:** Kanglei Zhang, Jiahui Li, Wenxuan Dong, Qing Huang, Xueru Wang, Kai Deng, Waseem Ali, Ruilong Song, Hui Zou, Di Ran, Gang Liu, Zongping Liu

**Affiliations:** 1College of Veterinary Medicine, Yangzhou University, Yangzhou 225009, China; dz120200021@stu.yzu.edu.cn (K.Z.); mz120211583@stu.yzu.edu.cn (J.L.); mx120221013@stu.yzu.edu.cn (Q.H.); dx120220199@stu.yzu.edu.cn (X.W.); mz120211581@stu.yzu.edu.cn (K.D.); 008301@yzu.edu.cn (W.A.); rlsong@yzu.edu.cn (R.S.); zouhui@yzu.edu.cn (H.Z.); 2Jiangsu Co-Innovation Center for Prevention and Control of Important Animal Infectious Diseases and Zoonoses, Yangzhou 225009, China; 3Joint International Research Laboratory of Agriculture and Agri-Product Safety, The Ministry of Education of China, Yangzhou University, Yangzhou 225009, China; 4College of Veterinary Medicine, Qingdao Agricultural University, Qingdao 266000, China; dx120190152@stu.yzu.edu.cn; 5College of Veterinary Medicine, Southwest University, Chongqing 400715, China; diran@uic.edu; 6College of Medicine, University of Illinois at Chicago, Chicago, IL 60607, USA; 7Department of Pathology and Laboratory Medicine, Tulane University School of Medicine, New Orleans, LA 70112, USA

**Keywords:** cadmium, luteolin, PARP-1, SIRT1, autophagic flux

## Abstract

Chickens are a major source of meat and eggs in human food and have significant economic value. Cadmium (Cd) is a common environmental pollutant that can contaminate feed and drinking water, leading to kidney injury in livestock and poultry, primarily by inducing the generation of free radicals. It is necessary to develop potential medicines to prevent and treat Cd-induced nephrotoxicity in poultry. Luteolin (Lut) is a natural flavonoid compound mainly extracted from peanut shells and has a variety of biological functions to defend against oxidative damage. In this study, we aimed to demonstrate whether Lut can alleviate kidney injury under Cd exposure and elucidate the underlying molecular mechanisms. Renal histopathology and cell morphology were observed. The indicators of renal function, oxidative stress, DNA damage and repair, NAD^+^ content, SIRT1 activity, and autophagy were analyzed. In vitro data showed that Cd exposure increased ROS levels and induced oxidative DNA damage and repair, as indicated by increased 8-OHdG content, increased γ-H2AX protein expression, and the over-activation of the DNA repair enzyme PARP-1. Cd exposure decreased NAD^+^ content and SIRT1 activity and increased LC3 II, ATG5, and particularly p62 protein expression. In addition, Cd-induced oxidative DNA damage resulted in PARP-1 over-activation, reduced SIRT1 activity, and autophagic flux blockade, as evidenced by reactive oxygen species scavenger NAC application. The inhibition of PARP-1 activation with the pharmacological inhibitor PJ34 restored NAD^+^ content and SIRT1 activity. The activation of SIRT1 with the pharmacological activator RSV reversed Cd-induced autophagic flux blockade and cell injury. In vivo data demonstrated that Cd treatment caused the microstructural disruption of renal tissues, reduced creatinine, and urea nitrogen clearance, raised MDA content, and decreased the activities or contents of antioxidants (GSH, T-SOD, CAT, and T-AOC). Cd treatment caused oxidative DNA damage and PARP-1 activation, decreased NAD^+^ content, decreased SIRT1 activity, and impaired autophagic flux. Notably, the dietary Lut supplement observably alleviated these alterations in chicken kidney tissues induced by Cd. In conclusion, the dietary Lut supplement alleviated Cd-induced chicken kidney injury through its potent antioxidant properties by relieving the oxidative DNA damage-activated PARP-1-mediated reduction in SIRT1 activity and repairing autophagic flux blockade.

## 1. Introduction

Cadmium (Cd) is a widely distributed environmental pollutant that seriously threatens global health and has attracted widespread attention from society. Cd is mainly released into the environment through human activities, such as mineral mining, metal smelting, fertilizer production, and waste disposal [1]. More seriously, the accumulation of Cd in water, soil, and sediments is still increasing [2,3]. The absorbed Cd accumulates in the body due to its low excretion rate and long biological half-life, eventually causing damage to various organs and tissues [4,5]. The kidney plays an important role in metabolic waste excretion and is also the primary target organ of Cd toxicity [6]. Renal tubular epithelial cells (RTECs) are very sensitive to Cd exposure. It has been reported that Cd-induced RTEC damage is irreversible, and continuous exposure to Cd can increase the risk of developing whole kidney damage [6]. Studies showed that Cd could cause kidney injury through various mechanisms, such as oxidative stress, apoptosis, autophagy, and inflammatory response [7,8,9]. However, the underlying mechanisms of Cd-induced kidney toxicity in chickens are not fully understood.

Oxidative stress is the key mechanism of Cd toxicity, which can oxidize biomolecules such as DNA to lose their biological function. As a DNA damage response protein, poly (ADP-ribose) polymerase 1 (PARP-1) is reported in all kinds of eukaryotic cells. It is activated during DNA damage reparation and maintains genome integrity [10]. Notably, PARP-1 activity requires consuming a large amount of nicotinamide adenine dinucleotide (NAD^+^). Studies have shown that PARP-1 plays a dual role in pro-survival or pro-death based on varying degrees of DNA damage [11,12]. Silent mating-type information regulation 2 homolog-1 (SIRT1) is a vital deacetylase that regulates inflammation, aging, mitochondrial biogenesis, and autophagy [13,14]. Like PARP-1, SIRT1 activity is strictly dependent on NAD^+^ levels. It has been reported that the excessive activation of PARP-1 causes massive NAD^+^ depletion, leading to decreased SIRT1 activity [15]. Conversely, SIRT1 inhibits PARP-1 activity via deacetylation and protects mouse hearts from PARP-1-mediated cell death [16]. However, the role of SIRT1 or PARP-1 and their crosstalk in Cd-induced chicken kidney injury is still unclear.

Autophagy is an evolutionarily conserved metabolic process that degrades cellular components for recycling through lysosomes. Autophagy is considered as a protective response to changes in the microenvironment caused by various stimuli. However, defective or excessive autophagy contributes to the occurrence and development of multiple diseases [17,18,19]. SIRT1 plays an essential role in the transcription of autophagy-related genes and maintains autophagic flux [20,21]. Other studies have reported that SIRT1-regulated autophagy is associated with PARP-1 activation [22,23]. In addition, SIRT1 is degraded via the autophagic pathway to maintain its homeostasis [24]. Whether SIRT1 protects kidney injury induced by Cd in chickens by regulating autophagy deserves further study.

Luteolin (Lut) is a natural flavonoid compound, and peanut shell is one of the important extraction sources. It plays a key role in the antioxidation, anti-inflammation, and regulation of autophagy and thus benefits both human and animal health [25,26]. Many studies have found that Lut effectively ameliorates heavy metal-induced pathological lesions in various organs by enhancing antioxidant enzyme activities and reversing mitochondrial dysfunction [27,28,29]. Moreover, Lut inhibits PARP-1 activation and apoptosis induced by a mixture of heavy metals via reducing reactive oxide species (ROS) levels [30]. Recent studies have reported that the hepatorenal protective effects of Lut involve the activation of SIRT1 and the repair of impaired autophagic flux [31,32,33]. However, whether Lut can mitigate Cd-induced oxidative damage in chicken kidneys and its underlying mechanisms remain unknown.

In the present study, we aim to clarify the crosstalk between PARP-1 and SIRT1 and their roles in the regulation of autophagy in kidney injury under Cd exposure in vitro. We further demonstrate whether Lut can alleviate Cd exposure-induced kidney injury in chickens and its underlying mechanism. This study identifies effective therapeutic targets and provides potential medicine options for preventing kidney injury in poultry.

## 2. Materials and Methods

### 2.1. Reagents and Chemicals

Lut was obtained from Yuanye Bio-Technology Co., Ltd. (YuanYe, Shanghai, China). Cadmium chloride (CdCl_2_, 202908), anti-p62 (P0067), anti-LC3 (L7543), and N-Acetyl-L-cysteine (NAC, 30498) were purchased from Sigma-Aldrich (Sigma, St. Louis, MO, USA). The PARP-1 inhibitor PJ34 (HY-13688A) and the SIRT1 activator resveratrol (RSV, HY-16561) were purchased from MedChemexpress CO., Ltd. (MCE, Shanghai, China). Anti-Beclin-1 (3495S), anti-β-actin (4970L), and anti-ATG5 (12994S) were purchased from (CST, Danvers, MA, USA). Anti-PARP-1(13371-1-AP) and anti-SIRT1 (13161-1-AP) were obtained from Wuhan Sanying Biotechnology Co., Ltd. (Proteintech, Wuhan, China). Anti-ac-H4K16 (ab109463) and anti-γ-H2AX (ab81299) were purchased from Abcam Trading Co., Ltd. (Abcom, Cambridge, UK). The ROS detection kit (S0033S), the NAD^+^/NADH assay kit (S0179), and the BCA protein assay kit (P0009, Beyotime, Shanghai, China) were purchased from Beyotime Biotech Inc. (Beyotime, Shanghai, China). The malondialdehyde (MDA) assay kit (A003-1-2), reduced glutathione (GSH) assay kit (A006-2-1), total antioxidant capacity assay kit (T-AOC, A015-2-1), total superoxide dismutase (T-SOD) assay kit (A001-1-2) and catalase (CAT) assay kit (A007-1-1) were purchased from Nanjing Jiancheng Bio-Engineering Institute Co., Ltd. (Nanjing, China). AffiniPure™ Goat Anti-Rabbit IgG (AB_2337913) and AffiniPure™ Goat Anti-Mouse IgG (AB_2338447) were obtained from Jackson ImmunoResearch Laboratories Inc. (Jackson, CST, MA, USA). Chicken 8-hydroxy-2-deoxyguanosine (8-OHdG) ELISA kit (ml059825) was obtained from Shanghai Enzyme-linked Biotechnology Co., Ltd. (mlbio, Shanghai, China).

### 2.2. Animals and Experimental Design

One-day-old suvian green shell layer chickens (n = 60) were purchased from Yangzhou Xianglong Livestock Development Co., Ltd. (Yangzhou, Jiangsu, China). The average body weight was (83.62 ± 2.20 g). All chickens were reared at a constant temperature and humidity in a well-ventilated animal house. Balanced commercial feed and tap water were provided to the experimental chickens ad libitum. All animal care procedures and protocols employed in this study were approved by the Animal Care and Ethics Committee of Yangzhou University (Approved No. SYXK [Su] 2017-0044). The dosage of 50 mg/L CdCl_2_ was chosen based on the previous studies but with slight modifications [34,35]. Previous studies have shown that the oral administration of 100 to 200 mg/kg Lut can alleviate heart and kidney injury in rats [36,37]. Our preliminary experiment found that oral treatment with 100, 150, and 200 mg/kg doses for 7 days had no effect on the growth and behavioral performance of chickens. Therefore, a moderate dose of 150 mg/kg was chosen to explore the protective effect of Lut on Cd-induced kidney injury in chickens. After 7 days of pre-feeding, the chickens were randomly and equally divided into 4 groups, with 15 chickens in each group. The chickens were grouped and treated as follows: the control/untreated group (were fed with basal diet continuously for the whole period), the Cd group (were pre-fed with basal diet for the first 7 days, and continued to be fed with basal diet supplemented with CdCl_2_ for 1 month from the 8th day), the Cd + Lut group (were pre-fed with basal diet supplemented with Lut for the first 7 days and continued to be fed basal diet supplemented with CdCl_2_ and Lut for 1 month from the 8th day), and the Lut group (were fed with basal diet supplemented with Lut continuously for the whole period). Both CdCl_2_ and Lut were fed to the chickens by mixing material through the addition of drinking water. The total time period of the experimental trial was 37 days. After the completion of the experimental trial, all chickens were euthanized and dissected to collect the kidneys immediately, which were subsequently stored with neutral tissue and an electron microscope fixative or in an −80 °C ultra-low temperature freezer.

### 2.3. Extraction, Culture and Treatment of Chicken RTECs

The chicken RTECs were isolated from suvian green shell layer chicken embryos by gauze filtration and collagenase digestion combined with differential centrifugation. The 9 to 12-day-old chicken embryos were sterilized and placed on a sterile operating table. Kidney tissue was isolated and then digested using collagenase. The digestion was terminated using a complete culture medium, followed by filtration using gauze and subsequent differential centrifugation to harvest RTECs. Finally, the resuspended cells were seeded in appropriate cell culture dishes according to experimental requirements.

The isolated chicken RTECs were cultured in a cell incubator (37 °C; 5% CO_2_) until the confluence rate reached about 60% before subsequent treatment. Cell experiments were grouped and treated as follows: (1) treatment with different concentrations of Cd was divided into 3 or 4 groups at 0 or 1.25, 2.5, and 5 μM CdCl_2_ for 12 h, respectively; (2) NAC and Cd treatment were divided into the following 4 groups: control/untreated group (ultrapure water), Cd group (ultrapure water + 5 μM CdCl_2_), Cd + NAC group (NAC pretreatment for 1 h, followed by NAC and CdCl_2_ co-treatment for 12 h), and NAC group (200 μM NAC); (3) PJ34 and Cd treatment were divided into the following 4 groups: control/untreated group (DMSO), Cd group (MDSO + 5 μM CdCl_2_), Cd + PJ34 group (pretreatment of PJ34 for 1 h, followed by PJ34 and CdCl_2_ co-treatment for 12 h), and PJ34 group (10 μM PJ34); (4) RSV and Cd treatment were divided into the following 4 groups: control/untreated group (DMSO), Cd group (DMSO + 5 μM CdCl_2_), Cd + RSV group (RSV pretreatment for 1 h, followed by RSV and CdCl_2_ co-treatment for 12 h), RSV group (10 μM RSV).

### 2.4. Observation of Histopathology and Ultrastructure

The histopathological observation of renal tissues using hematoxylin and eosin (H&E) staining was performed as previously described in our studies [38]. Fresh renal cortex was isolated and immediately fixed in a neutral tissue fixative and subsequently dehydrated in alcohol, clear in xylene, and embedded in paraffin wax. The embedded tissues were sliced into sections of about 5–8 μm in thickness, flattened on a slide, and dried in an incubator. Tissue sections were deparaffinized in different concentrations of alcohol (75–100%) and stained with H&E staining. The prepared sections were observed under the bright field of a phase-contrast microscope (Leica, Wetzlar, Germany).

Ultrastructure observation was described in our previous studies [38]. Briefly, kidney tissues were fixed in 2.5% glutaraldehyde, washed overnight with a buffer, fixed in osmium tetroxide (2%), dehydrated in ethanol and propylene oxide, embedded, sectioned, and then stained with lead citrate combined with uranyl acetate. Finally, the prepared sections were visualized by transmission electron microscopy (TEM) (HT7700; Hitachi, Tokyo, Japan).

### 2.5. Detection of Biochemical Parameters

Blood was collected by exsanguinating the jugular vein of the chicken, which was placed at 37 °C for 30 min, followed by centrifugation at 2500 rpm for 10 min to separate the serum. The levels of creatinine (CREA), glucose (Glu), and uric acid (UA), which were related to renal function, were detected using an AU5800 automatic biochemical analyzer (Beckman Coulter, Brea, CA, USA).

### 2.6. Determination of Metal Concentrations

Renal tissue samples were placed in an oven for 48–72 h to ensure the dehydration of the samples. The dried kidney tissue was weighed and subsequently completely digested with nitric acid using a microwave digester (Intertek, London, UK). The digested solution was adjusted to the same volume. The relevant metal element content was determined using a flame atomic absorption spectrophotometer (FAAS) (PerkinElmer, Waltham, MA, USA). The final statistical results were expressed as the ratio of the weight of the element (μg) to the weight of the dried kidney tissue (g).

### 2.7. Measurement of ROS and Oxidative Stress Indexes

The activities of CAT and T-SOD, the content of GSH and MDA, and T-AOC were measured using commercially available kits. The specific procedure was referred to in the previous study [38]. The protein concentration of the samples was determined using the BCA protein assay kit to normalize the MDA content as well as the content or activity of the above-mentioned antioxidants.

For cellular ROS detection, chicken RTECs were incubated using a DCFH-DA working solution for 20 min in a 37 °C incubator and subsequently digested with trypsin and centrifuged. After cells were resuspended and washed in a serum-free culture medium, ROS levels were analyzed using flow cytometry (BD, Franklin Lakes, NJ, USA). DCF fluorescence intensities were analyzed using FlowJo 7.6.1 software (BD, Franklin Lakes, NJ, USA).

### 2.8. Detection of NAD^+^ and NADH Contents

The tissue or cell homogenate was centrifuged at 12,000× *g* in a freezing centrifuge to separate the supernatant, which was the sample to be tested. In total, 50 to 100 μL of the sample was treated at 60 °C for 30 min to decompose NAD^+^ (for the detection of NADH), whereas the remaining sample was left untreated (for the detection of total NAD^+^ and NADH). In total, 20 μL of the above two groups of samples was transferred to a 96-well plate, and blank wells and standard wells (NAD^+^ standard solution) were set up simultaneously and then incubated with an alcohol dehydrogenase solution at 37 °C for 10 min in the dark. The chromogenic solution was added, and incubation continued for 30 min. The absorbance value was detected using a spectrophotometer at 450 nm, and the NAD^+^ and NADH contents were calculated according to the standard curve. In addition, the protein concentration of each sample was determined using the BCA protein assay kit to normalize the NAD^+^ or NADH content.

### 2.9. Determination of 8-OHdG Content

Homogenates of cells and tissues were transferred to ELISA-specific slats. The horseradish peroxidase-labeled assay antigen was added to each well and incubated at 37 °C for 60 min. After washing, a mixture of substrates A and B were supplemented and incubated at 37 °C for 15 min. The termination solution was subsequently added, and the absorbance value was detected using a spectrophotometer at 450 nm. 8-OHdG content was calculated from standard curves, and protein concentrations were used to normalize 8-OHdG content.

### 2.10. Immunohistochemistry (IHC)

Paraffin-embedded tissue sections were deparaffinized in xylene and then hydrated in a gradient of absolute ethanol and ultrapure water. Tissue sections were processed with citrate buffer for antigen repair for 30 min, treated with 3% hydrogen peroxide for 15 min, blocked with goat serum, incubated overnight with the LC3 antibody at 4 °C, applied a secondary antibody at room temperature for 2 h, before subsequently staining with DAB. Sections treated above continued to be counterstained with hematoxylin, differentiated in 1% hydrochloric ethanol, and dehydrated in ethanol and xylene. Finally, visualization was performed using a phase-contrast microscope.

### 2.11. TdT-Mediated dUTP Nick-End Labeling (TUNEL) Staining

In brief, tissue sections were deparaffinized with xylene, hydrated with gradient alcohol, washed with PBS, and then permeabilized with Proteinase K for 10–30 min at room temperature. This section was reacted for 1 h at 37 °C in a wet box using the TUNEL reaction mix, followed by three washes with PBS. Sections were incubated with Streptavidin labeled HRP for 30 min, followed by three washes with PBS. Sections were reacted with the DAB mixture for 10 min, then counterstained with hematoxylin, dehydrated with gradient alcohol, and cleared with xylene. Finally, the sections were observed using a phase-contrast microscope.

### 2.12. Western Blotting Analysis

Cell and renal tissue protein samples were extracted using the RIPA lysates buffer (20101ES60, NCM, Suzhou, China), and protein concentrations were normalized using the BCA protein assay kit. Samples were added with the SDS buffer and boiled for 8–15 min to denature the proteins. Protein samples were separated in 8–12% SDS-PAGE gel by electrophoresis and subsequently transferred to the 0.22 or 0.45 μm PVDF membrane (Merck Millipore, Darmstadt, HE, GER). Membranes were blocked with 5% nonfat milk, incubated with primary antibody at 4 °C overnight, and incubated with secondary antibody for 1–2 h at room temperature. Finally, Western blotting images were visualized using an enhanced chemiluminescence reagent (NCM, Suzhou, China) via the Tanon imaging system. Image J 1.42q software (NIH, Bethesda, MD, USA) was utilized to analyze protein expression, and the β-actin was used to normalize the protein expression.

### 2.13. Statistical Analysis

The results were analyzed using software IBM SPSS Statistics 19 statistical software (IBM, Armonk, NY, USA). Dates with normal distribution were presented with the mean ± standard error of the mean (SEM). One-way analysis of variance was used for comparison between the groups, and the least significant difference (LSD) was used for post hoc analysis. *p* < 0.05 was considered statistically significant. Graphs were drafted using GraphPad Prism 6 software (San Diego, CA, USA).

## 3. Results

### 3.1. Cd-Induced Oxidative DNA Damage, PARP-1 Over-Activation, Decreased SIRT1 Activity, and Autophagic Flux Blockade in Chicken RTECs

Chicken RTECs were treated with different concentrations of Cd for 12 h to study the underlying mechanisms of Cd on renal tubular epithelial cell injury. Bright-field observation showed that the cells were standard ovoid-shaped with high cell confluency in the control group, but low cell confluency and more shrinking cells were observed after Cd exposure (Figure 1A). The flow cytometry results showed that the level of ROS increased significantly (*p* < 0.01) in a concentration-dependent manner after Cd exposure (Figure 1B). 8-OHdG and γ-H2AX are widely recognized biomarkers of oxidative DNA damage. As shown in Figure 1C–E, compared with the control group, the 8-OHdG content and the protein expression of γ-H2AX and PARP-1 were observably increased after Cd treatment (*p* < 0.05 or *p* < 0.01). In addition, Cd exposure significantly decreased the NAD^+^ content and the ratio of NAD^+^/NADH (*p* < 0.05 or *p* < 0.01) (Figure 1F,G). Cd exposure dramatically decreased the protein expression of SIRT1 (*p* < 0.01) and increased the protein expression of ac-H4K16 (*p* < 0.05 or *p* < 0.01), respectively (Figure 1H,I). Furthermore, Cd exposure remarkably increased LC3 II, ATG5, and p62 protein expression in particular (*p* < 0.05 or *p* < 0.01) (Figure 1J). These results indicate that Cd caused oxidative DNA damage, PARP-1 over-activation, SIRT1 activity reduction, and autophagic flux blockade in chicken RTECs.

### 3.2. Alleviating Oxidative Stress Attenuated Cd-Induced PARP-1 Over-Activation, SIRT1 Inhibition, and Autophagic Flux Blocking in Chicken RTECs

NAC, a recognized potent ROS scavenger, is often used to determine whether oxidative stress is involved in risk factor-induced biotoxicity. The effects of oxidative stress on Cd-induced renal tubular epithelial cell injury were investigated by pretreatment with NAC and subsequent co-treatment with Cd for 12 h. NAC treatment improved the morphological abnormalities and low cell confluency induced by Cd (Figure 2A). Compared with the Cd-treated group, NAC treatment significantly decreased (*p* < 0.01) the ROS level (Figure 2B). We further investigated the effect of NAC on DNA damage and the autophagic flux blockade caused by Cd. NAC treatment observably reduced the 8-OHdG content (*p* < 0.01) and the protein expression of γ-H2AX and PARP-1 (*p* < 0.05 or *p* < 0.01) (Figure 2C–E). The NAC treatment dramatically reversed the reduction in NAD^+^ content (*p* < 0.01) and the NAD^+^/NADH ratio (*p* < 0.01) induced by Cd exposure (Figure 2F,G). Western blotting analysis showed that NAC treatment significantly increased SIRT1 protein expression (*p* < 0.01) and decreased ac-H4K16 protein expression (*p* < 0.05) (Figure 2H,I). Furthermore, NAC treatment observably reduced LC3 II and ATG5 protein expression (*p* < 0.05 or *p* < 0.01) (Figure 2J,K). Notably, NAC significantly decreased the Cd-induced increase of p62 protein expression (*p* < 0.01) (Figure 2J,K). These results suggest that the mitigation of oxidative stress using the antioxidant NAC effectively alleviated Cd-induced oxidative DNA damage, PARP-1 over-activation, and autophagic flux blockade in chicken RTECs.

### 3.3. Inhibition of PARP-1 Restored SIRT1 Activity and Autophagic Flux in Chicken RTECs

Next, chicken RTECs were pretreated with the PARP-1 inhibitor PJ34 followed by co-treatment with Cd for 12 h to study the effect of PARP-1 on SIRT1 activity and autophagic flux. The result showed that PJ34 treatment noticeably alleviated the cell morphological changes and reduced cell confluency compared with the Cd-treated group (Figure 3A). PJ34 treatment significantly weakened ROS accumulation (*p* < 0.05), inhibited PARP-1 protein over-activation (*p* < 0.01), and reduced γ-H2AX protein expression (*p* < 0.01) (Figure 3B,E,F). The reduction in NAD^+^ content and the NAD^+^/NADH ratio, decreased SIRT1 protein, and increased ac-H4K16 protein induced by Cd was remarkably reversed by PJ34 treatment (*p* < 0.05 or *p* < 0.01) (Figure 3C,D,G,H). Furthermore, PJ34 treatment increased the expression of the LC3 II and ATG5 proteins but decreased the expression of the p62 protein (*p* < 0.05) (Figure 3I,J).

In addition, chicken RTECs were pretreated with the SIRT1 activator RSV followed by co-treatment with Cd for 12 h to explore the regulatory role of SIRT1 on autophagy. Compared with the Cd-treated group, RSV significantly increased LC3 II protein expression (*p* < 0.05) but significantly reduced p62 protein expression (*p* < 0.01) (Figure 4A,B). Unsurprisingly, RSV treatment significantly reduced the ROS and 8-OHdG content (*p* < 0.05), as well as alleviating morphological changes (Figure 4C–E). These results indicated that the inhibition of PARP-1 restored SIRT1 activity reduction and the autophagic flux blockade caused by Cd in chicken RTECs.

### 3.4. Lut Alleviates Cd-Induced Kidney Injury

We explored the protective effect of Lut against Cd-induced renal injury. The results showed that Cd content in renal tissues and serum significantly increased (*p* < 0.01) after Cd exposure but did not change considerably (*p* > 0.05) in the Cd + Lut-treated group (Figure 5A and Appendix A). Cd exposure significantly reduced body weight (*p* < 0.05) compared to the control group, while the Lut supplement significantly alleviated the body weight loss caused by Cd (*p* < 0.05) (Figure 5B). However, Cd or Lut treatment had no significant effect on kidney weight (*p* > 0.05) (Figure 5C). Cd exposure significantly increased the kidney coefficient (*p* < 0.05), whereas the Lut supplement had a lower kidney coefficient than the Cd-treated group (*p* < 0.05) (Figure 5D). The histopathological evaluation by H&E staining showed that Cd exposure caused the irregular arrangement of renal tubular cells, swelling, and vacuolization of epithelial cells (Black arrows). However, the Lut supplement significantly alleviated Cd-induced tissue lesions (Figure 5E). Moreover, serum biochemical results showed that CREA, UA, and Glu levels increased dramatically (*p* < 0.05) after Cd exposure and were clearly reversed (*p* < 0.05 or *p* < 0.01) in the Cd + Lut-treated group (Figure 5F–H). These results suggest that the Lut supplement considerably ameliorated Cd-induced kidney injury mainly by repairing the renal structure and maintaining normal renal function.

### 3.5. Lut Alleviates Cd-Induced Oxidative Stress

We explored whether the Lut supplement could alleviate Cd-induced oxidative stress in renal tissue. The TEM result showed an irregular arrangement of mitochondrial cristae, even fragmentation or loss, and outer membrane disruption in the Cd-treated group. The Lut supplement significantly alleviated the disruption of the ultrastructure of mitochondria (Figure 6A). The MDA content and antioxidant indicator were measured to assess oxidative stress status in the kidneys. As shown in Figure 6B, compared with the Cd-treated group, the Lut supplement noticeably decreased renal MDA accumulation (*p* < 0.01) and GSH elevation (*p* < 0.05), and observably improved the activity of T-SOD, T-AOC, and CAT (*p* < 0.05) (Figure 6C–F). Trace element results showed that Cd exposure significantly increased the contents of zinc (Zn) and copper (Cu) compared with the control group (*p* < 0.01). The Lut supplement considerably reduced Cd-induced increases in the Zn content but did not affect the Cu content (*p* < 0.05 or *p* > 0.05) (Figure 6G,H). However, no significant difference showed in iron (Fe) and selenium (Se) content among the groups (*p* > 0.05) (Figure 6I,J). These results suggest that the Lut supplement remarkably ameliorated renal oxidative stress under Cd exposure by reducing the accumulation of MDA and enhancing the activity of antioxidant enzymes.

### 3.6. Lut Attenuates Cd-Induced DNA Damage and PARP-1 Activation

We studied the effects of Lut supplement and/or Cd exposure on DNA damage and PARP-1 activation. As shown in Figure 7A, the morphology of the nucleus was regularly round, and the nuclear membrane was intact in the control group, while the nuclear morphology was deformed in the Cd-treated group. The Lut supplement significantly alleviated the abnormal nuclear morphology caused by Cd. The number of TUNEL-positive cells was dramatically increased after Cd exposure and significantly decreased in the Cd + Lut-treated group (Figure 7B). As shown in Figure 7C–E, the Lut supplement significantly reduced the increases in 8-OHdG (*p* < 0.01), γ-H2AX (*p* < 0.05), and PARP-1 (*p* < 0.05) induced by Cd. These results demonstrate that Lut significantly alleviates Cd-induced DNA oxidative damage and PARP-1 activation in chicken kidney tissue.

### 3.7. Lut Restores Cd-Induced SIRT1 Inactivity and Autophagic Flux Blockade

We explored the effects of the Lut supplement and/or Cd exposure on SIRT1 activity and autophagy in the kidney. As shown in Figure 8A,B, Cd exposure significantly decreased the NAD^+^ content (*p* < 0.05) and the ratio of NAD^+^/NADH (*p* < 0.05), which were observably improved by the Lut supplement (*p* < 0.05). Cd exposure greatly reduced the SIRT1 protein level (*p* < 0.05) but increased the ac-H4K16 protein expression (*p* < 0.01), and these changes were reversed considerably (*p* < 0.05) by the Lut supplement (Figure 8C,D).

Next, the TEM results show that the Lut supplement reduced the number of autophagosomes or autophagic vesicles elevated by Cd (Figure 9A). The IHC results showed that Cd exposure significantly increased LC3 protein expression, while this increase was weakened by the Lut supplement (Figure 9B). In addition, Western blotting analysis confirmed that the protein expression levels of p62, LC3 II, ATG5, and Beclin-1 were significantly increased (*p* < 0.01) after Cd exposure, while they were decreased considerably (*p* < 0.05 or *p* < 0.01) by the Lut supplement (Figure 9C,D). These results illustrate that the Lut supplement repaired the SIRT1 inactivity and autophagic flux blockade induced by Cd in chicken kidney tissue.

## 4. Discussion

Cd is a widely distributed environmental and occupational pollutant that has adverse effects on human and animal health and has been a global public health problem. Cd exposure is closely related to the development of various kidney diseases, including diabetes, interstitial renal fibrosis, glomerulosclerosis, and even renal cell carcinoma [39]. The exploration of effective therapeutic interventions is necessary to treat Cd-induced nephrotoxicity in animals and humans. Recently, Lut, a natural flavonoid, attracted much attention due to its various pharmacological activities. Studies have reported that Lut could protect organs such as the kidneys and liver from damage caused by heavy metals or chemicals [32,33]. However, the therapeutic effects of Lut on Cd-induced kidney injury in chickens have not been fully studied. Therefore, our study aimed to explore the protective effect of Lut against Cd-induced nephrotoxicity and elucidate the underlying molecular mechanisms.

Numerous studies have demonstrated that Cd causes renal pathological changes in poultry [40,41,42,43]. Lut can effectively improve kidney injury induced by various stimuli [31,32,44]. In addition, Lut also plays a protective role in resisting heavy metal poisoning [45,46]. Similar to previous studies, our results demonstrated that Cd exposure caused structural and functional abnormalities in the kidney, and Lut could significantly alleviate these changes. Studies have found that Lut reduces the accumulation of heavy metals in organs by chelating or increasing the excretion of heavy metals, including lead or mercury in the urine thereby alleviating the damage to the kidney and testis in rats [27,31]. Our study found that Cd exposure noticeably increased Cd contents in serum and renal tissues. However, Lut did not change its levels in the kidneys and serum, suggesting that the renoprotective effect of Lut is independent of reducing renal Cd accumulation. In addition to its chelating ability, the protective effect of Lut against organ damage is closely related to its potent antioxidant properties [47,48,49]. More and more studies have confirmed that oxidative stress is a crucial mechanism of Cd-induced nephrotoxicity [50,51]. In addition, studies have reported how Cd exposure causes an increase in lipid peroxidation products, a reduction in antioxidant enzyme capacity, and severe oxidative kidney injury in poultry [34,42,43]. Similar to previous reports, we found that Cd exposure caused the disruption or loss of mitochondrial crista and disruption of the outer membrane, increasing the MDA content and reducing antioxidant capacity or activity. In vitro studies also found that Cd exposure could significantly destroy cell morphology and elevate ROS content. ROS scavenger NAC treatment significantly reduced the ROS content and alleviated the morphological changes induced by Cd in RTECs. Expectedly, Lut also considerably improved Cd-induced oxidative stress, as indicated by the repair of mitochondrial ultrastructure damage, enhancing antioxidant enzyme activity, and reducing MDA content. The dyshomeostasis of trace elements in the body can promote oxidative damage and contribute to Cd toxicity [52]. Trace elements, including Zn, Cu, and Fe, are essential for maintaining normal cell structure and function. Cd can displace Zn and Cu and reduce antioxidant capacity [53,54]. The exploration of Cd could increase Cu and Zn levels but decrease Fe levels in chicken kidneys [34]. We found that Cd exposure caused an increase in Cu and Zn contents, and the Lut supplement reduced Zn content but did not affect Cu content. However, the contents of Se and Fe in renal tissue in each treatment group did not change significantly. The invariableness in Fe content may be related to the transfer of stored Fe from the liver to the kidney after Cd exposure [55]. Cu/Zn is a metal prosthetic group of the antioxidant enzyme SOD, and the activity of Cu/Zn-SOD was further detected to better evaluate the protective effect of Lut on Cd-induced oxidative renal injury. Our results demonstrate that Cd exposure induces oxidative kidney injury, while Lut enhances the antioxidant capacity to alleviate Cd-induced nephrotoxicity.

Cd can induce oxidative DNA damage in rat kidneys and RTECs due to excess ROS production [12,56]. Recently, studies reported that Cd exposure causes oxidative stress in duck testes and broiler liver, which leads to DNA damage and the activation of autophagy or apoptosis [57,58]. Our study found that Cd caused nuclear deformation and an increase in the number of TUNEL-positive cells in renal tissues. In addition, Cd exposure increased 8-OHdG content and the γ-H2AX protein level, indicating that Cd exposure caused oxidative DNA damage in chicken kidneys. Not surprisingly, Lut reversed the oxidative DNA damage induced by Cd exposure in renal tissue due to its enhanced antioxidant capacity. It is reported that Lut has a protective effect on ROS-induced DNA damage, which is more pronounced than quercetin and kaempferol [59]. PARP-1 plays an important role in maintaining DNA repair and genome integrity. Notably, PARP-1 activation requires the consumption of NAD^+^ [60]. Cd causes oxidative stress and DNA fragmentation accompanied by PARP-1 activation and apoptosis in mammals and poultry [11,61]. Similar to previous studies, we also found that Cd caused a decrease in PARP-1 protein expression, NAD^+^ content, and the NAD^+^/NADH ratio. NAC could significantly alleviate the oxidative DNA damage induced by Cd, reduce PARP-1 activation, and increase NAD^+^ content and the NAD^+^/NADH ratio. Thus, DNA damage takes the primary responsibility for PARP-1 activation induced by Cd. Like NAC, Lut alleviated Cd-induced changes in PARP-1 protein expression and NAD^+^ content. Studies have found that the over-activation of PARP-1 induced by high concentrations of Cd promotes cell death, and conversely, moderate PARP-1 activation ameliorates cell damage induced by low concentrations of Cd [12,62]. Our study found that the inhibition of PARP-1 activation using PJ34 alleviated Cd-induced cell damage, suggesting that Cd-induced PARP-1 over-activation is detrimental to cell survival. In addition, PJ34 also reduced the expression of the γ-H2AX protein, which may be related to the alleviation of the ROS level induced by Cd. So, Lut protects the kidney from Cd toxicity by reversing oxidative DNA damage and PARP-1 over-activation.

SIRT1 is a deacetylase essential in regulating oxidative stress, inflammation, and autophagy by modifying downstream target proteins. In particular, SIRT1 activity also depends on the NAD^+^ content. The crosstalk between SIRT1 and PARP-1 and their roles in ultimate cell fate determination has rarely been investigated. One study indicated that the activation of PARP-1 reduces SIRT1 activity, which may involve a large amount of NAD^+^ consumption [63]. The inhibition of PARP-1 increases NAD^+^ content and SIRT1 activity [64,65]. However, SIRT1 can inhibit PAPR-1 activity by its deacetylation [16]. Studies have found that the SIRT1 protein was reduced in the Cd-induced rat poisoning model [56,66,67]. Cd exposure also decreases chicken renal tissues’ *SIRT1* and *SIRT3* gene expressions [34]. As expected, Cd exposure reduced the expression of the SIRT1 protein and increased the expression of its target protein ac-H4K16 both in vivo and in vitro. The Lut supplement reversed the changes in SIRT1 and ac-H4K16 proteins induced by Cd. Similarly, the results of NAC treatment were consistent with the Lut supplement. To further elucidate the role of PARP-1 activation on SIRT1 in Cd-induced chicken kidney injury, the inhibition of PARP-1 using PJ34 significantly upregulated the Cd-induced reduction in the NAD^+^ content and NAD^+^/NADH ratio and increased the reduction in the SIRT1 protein. Our results demonstrate that Lut restores SIRT1 activity by inhibiting the over-activation of PARP-1 induced by Cd.

Autophagy is a metabolic process that relies on lysosomes to degrade and reuse cellular components. Autophagy is closely related to Cd-induced nephrotoxicity in poultry. Cd upregulates the protein expression of LC3 II, ATG5, and Beclin-1 and reduces the expression level of the autophagy adaptor protein p62 in duck kidneys [68,69]. Recent reports have shown that Cd causes autophagic flux blockade and lipid droplet accumulation, leading to liver injury in ducks [70]. In our study, Cd exposure increased the autophagosome number in chicken kidney tissues. Meanwhile, Cd increased the protein expression of LC3 II, ATG5, or Beclin-1, as well as p62 in vitro and in vivo. ATG5 and Beclin-1 proteins are responsible for forming an autophagic isolation membrane [71]. Studies have found that Cd impairs autophagic flux by interfering with the degradation function of lysosomes and blocking the fusion of autophagosomes and lysosomes, manifesting as autophagosome accumulation and the increase in p62 protein expression [72]. Our results suggest that Cd impaired autophagic flux without affecting autophagosome formation, which resulted in the accumulation of a large number of autophagosomes. Consistent with our findings, Cd leads to autophagosome accumulation-dependent apoptosis in neuronal cells by activating AKT-mediated autophagic flux blockade [17,73]. A similar mechanism has been reported in cisplatin-induced nephrotoxicity [74]. It is necessary to further elucidate the role of autophagy in Cd-induced kidney injury in chickens by regulating autophagy using autophagy modulators, such as the activator rapamycin or the inhibitor 3-MA. In addition, further studies are also needed to clarify the underlying mechanisms of Cd-induced autophagic flux blockade in chicken kidneys. Notably, Lut significantly reversed Cd-induced autophagy-related protein expression and autophagosome accumulation. Similarly, NAC treatment downregulated the expression of autophagy-related proteins induced by Cd. Thus, the regulation of autophagy by Lut may be related to its antioxidant activity. In addition, it has been reported that competitive NAD^+^ depletion due to PARP-1 activation induced by DNA damage causes decreased autophagy, which is mediated by SIRT1 inactivation [22,75]. Moreover, the inhibition of PARP-1 further increased the protein expression of ATG5 and LC3 II and decreased p62 protein expression. It has been reported that activation of SIRT1 enhances autophagic flux in vivo, thereby protecting osteoporosis or alleviating hepatic steatosis [76,77]. We found that SIRT1 activation using RSV attenuated cell damage inhibited ROS production, increased LC3 II protein expression, and decreased p62 expression, indicating that Lut ameliorates autophagic flux blockade through PAPR-1 inhibition and SIRT1 activation to alleviate kidney injury in chickens.

## 5. Conclusions

Cd induces oxidative DNA damage and the over-activation of PARP-1, resulting in reduced SIRT1 activity and the blockade of autophagic flux in chicken kidney tissues and RTECs. The dietary supplementation of Lut protects the chicken kidney against Cd nephrotoxicity by repairing autophagic flux blockade via ameliorating oxidative DNA damage-dependent PARP-1 over-activation and SIRT1 activity reduction.

## Figures and Tables

**Figure 1 antioxidants-13-00525-f001:**
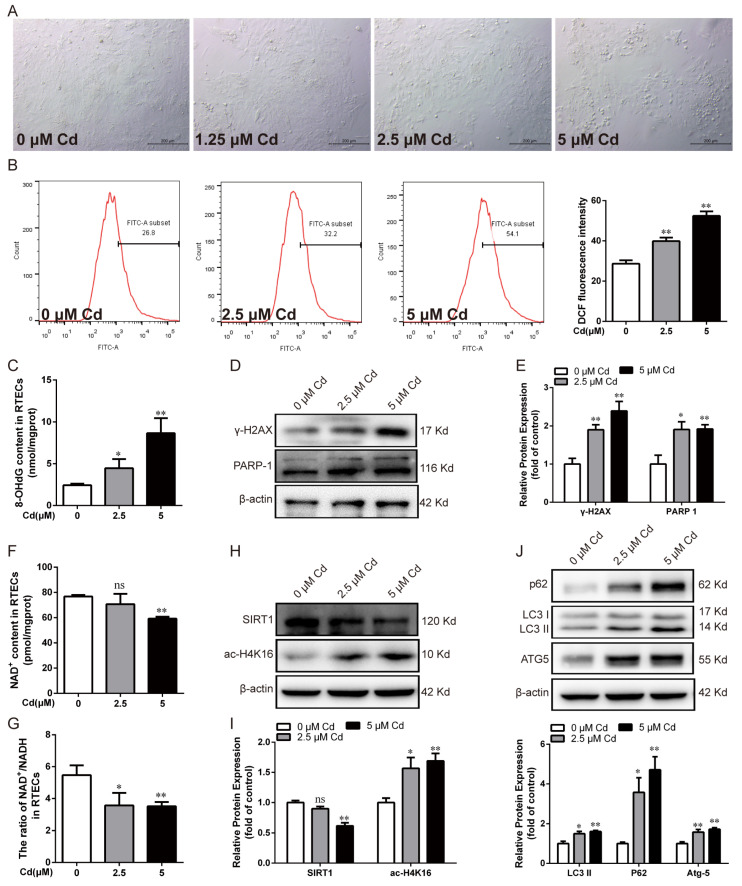
Effect of Cd on oxidative DNA damage, SIRT1 activity, and autophagy in chicken RTECs. (**A**) Cell morphology was observed by bright field under a phase-contrast microscope; scale bar: 200 μm. (**B**) The ROS content was detected by flow cytometry using DCFH-DA fluorescent probe staining. (**C**) The content of 8-OHdG was detected by ELISA. (**D**,**E**) Western blotting images and quantitative analysis of γ-H2AX and PARP-1 proteins. (**F**,**G**) Determination of NAD^+^ content and NAD^+^/NADH ratio using commercial kits. (**H**,**I**) Western blotting images and quantitative analysis of SIRT1 and ac-H4K16 proteins. (**J**) Western blotting images and quantitative analysis of autophagy-related proteins p62, LC3 II, and ATG5. Each experiment was duplicated at least three times. (ns: *p* ≥ 0.05; *: *p* < 0.05, **: *p* < 0.01).

**Figure 2 antioxidants-13-00525-f002:**
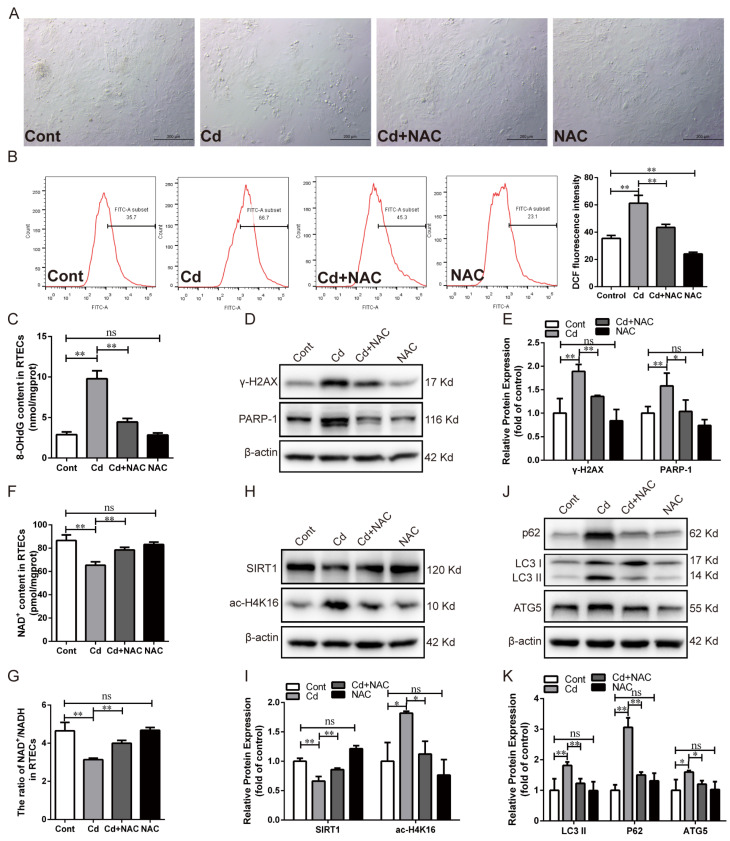
Effect of NAC on Cd-induced oxidative DNA damage, SIRT1 inhibition, and autophagic flux blockade in chicken RTECs. (**A**) Cell morphology was observed by bright field under a phase-contrast microscope; scale bar: 200 μm. (**B**) The ROS content was detected by flow cytometry using DCFH-DA fluorescent probe staining. (**C**) The content of 8-OHdG was detected by ELISA. (**D**,**E**) Western blotting images and quantitative analysis of γ-H2AX and PARP-1 proteins. (**F**,**G**) NAD^+^ content and NAD^+^/NADH ratio in cells. (**H**,**I**) Western blotting images and quantitative analysis of SIRT1 and ac-H4K16 proteins. (**J**,**K**) Western blotting images and quantitative analysis of autophagy-related proteins p62, LC3 II, and ATG5. Each experiment was duplicated at least three times. (ns: *p* ≥ 0.05; *: *p* < 0.05, **: *p* < 0.01).

**Figure 3 antioxidants-13-00525-f003:**
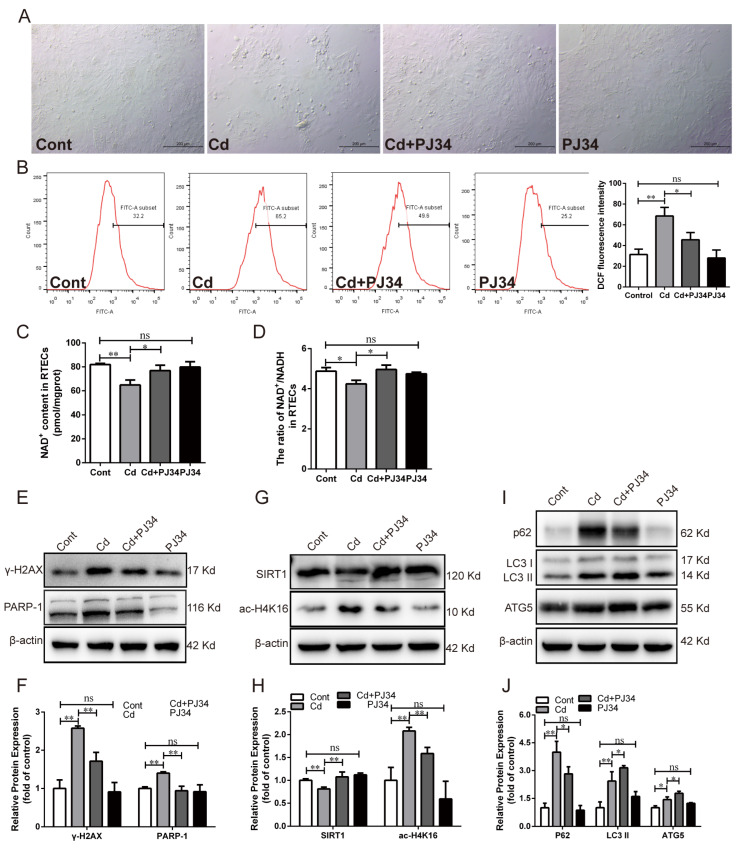
Effect of PARP-1 inhibition with PJ34 on Cd-induced SIRT1 activity reduction and autophagic flux blockade in chicken RTECs. (**A**) Bright-field observation of cell morphology under a phase-contrast microscope; scale bar: 200 μm. (**B**) The ROS content was detected by flow cytometry using DCFH-DA fluorescent probe staining. (**C**,**D**) NAD^+^ content and NAD^+^/NADH ratio in cells. (**E**,**F**) Western blotting images and quantitative analysis of γ-H2AX and PARP-1 proteins. (**G**,**H**) Western blotting images and quantitative analysis of SIRT1 and ac-H4K16 proteins. (**I**,**J**) Western blotting images and quantitative analysis of autophagy-related proteins p62 and LC3 II. Each experiment was duplicated at least three times. (ns: *p* ≥ 0.05; *: *p* < 0.05, **: *p* < 0.01).

**Figure 4 antioxidants-13-00525-f004:**
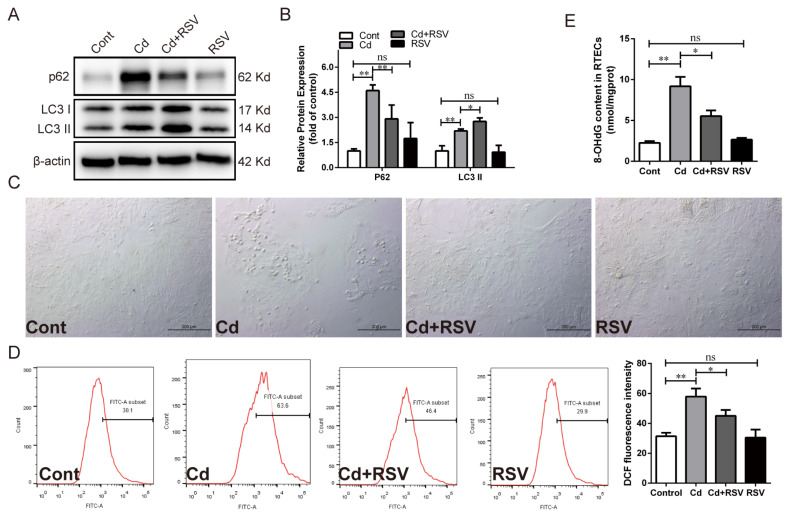
Effect of activating SIRT1 using RSV on Cd-induced autophagic flux blockade in chicken RTECs. (**A**,**B**) Western blotting images and quantitative analysis of autophagy-related proteins p62 and LC3 II. (**C**) Cell morphology was observed by bright field under a phase-contrast microscope; scale bar: 200 μm. (**D**) The ROS content was detected by flow cytometry using DCFH-DA fluorescent probe staining. (**E**) The content of 8-OHdG was detected by ELISA. Each experiment was duplicated at least three times. (ns: *p* ≥ 0.05; *: *p* < 0.05, **: *p* < 0.01).

**Figure 5 antioxidants-13-00525-f005:**
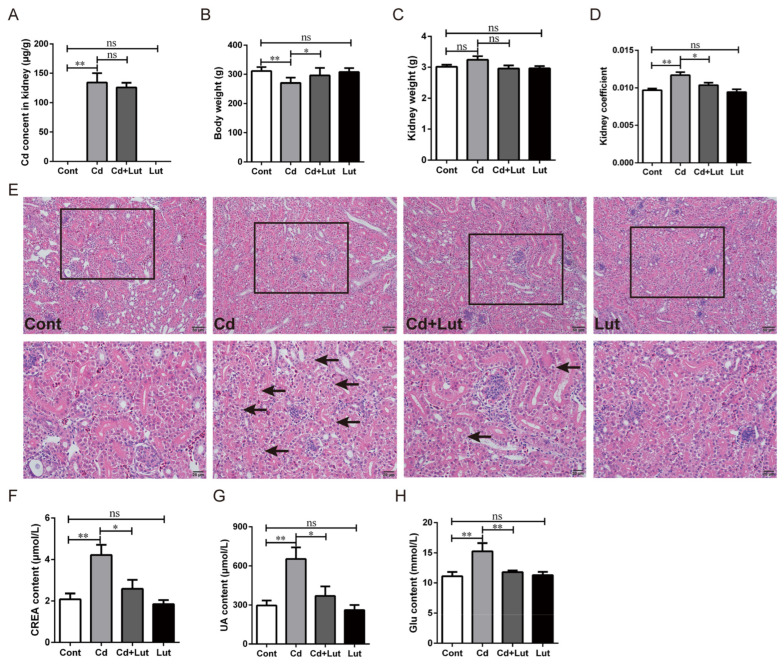
Effect of Lut and/or Cd on renal histopathology and renal function. (**A**) The Cd content in renal tissue was detected by FAAS. Statistical analysis of (**B**) body weight, (**C**) kidney weight, and (**D**) kidney coefficient. (**E**) Renal histopathological observation using H&E staining; scale bar (The black arrow indicates the abnormal morphology of the renal tubular, which is characterized by irregular arrangement of renal tubular cells, swelling and vacuolation of the epithelial cells). 50 μm or 20 μm. The contents of (**F**) CREA, (**G**) UA, and (**H**) Glu in serum were detected by an automatic biochemical analyzer. Each experiment was duplicated at least three times. (ns: *p* ≥ 0.05; *: *p* < 0.05, **: *p* < 0.01).

**Figure 6 antioxidants-13-00525-f006:**
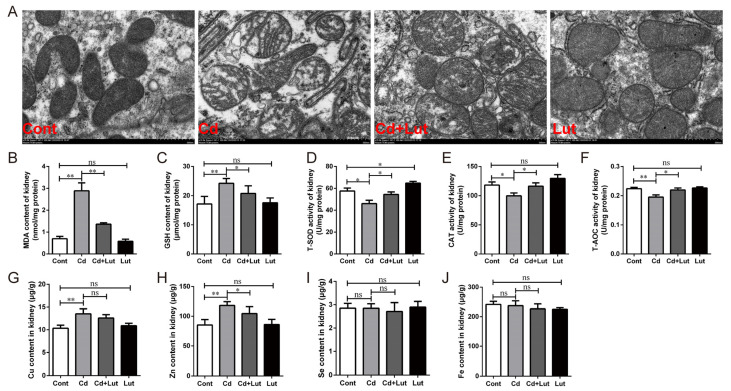
Effects of Lut and/or Cd on oxidative stress in chicken kidneys. (**A**) TEM observed mitochondrial ultrastructure; scale bar: 500 nm. (**B**) MDA content, (**C**) GSH content, (**D**) T-SOD activity, (**E**) CAT activity, and (**F**) T-AOC in renal tissues. The contents of (**G**) Cu, (**H**) Zn, (**I**) Se, and (**J**) Fe in renal tissues were measured by FAAS. Each experiment was duplicated at least three times. (ns: *p* ≥ 0.05; *: *p* < 0.05, **: *p* < 0.01).

**Figure 7 antioxidants-13-00525-f007:**
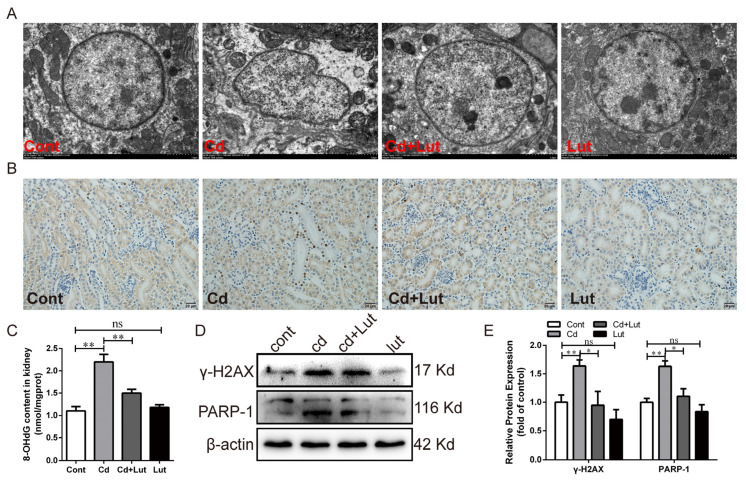
Effect of Lut and/or Cd on oxidative DNA damage and repair in chicken kidneys. (**A**) TEM observed nuclear morphology; scale bar: 1.0 μm. (**B**) DNA damage level detected by TUNEL staining; scale bar: 20 μm. (**C**) The content of 8-OHdG was detected by ELISA. (**D**,**E**) Western blotting images and quantitative analysis of γ-H2AX and PARP-1 proteins. Each experiment was duplicated at least three times. (ns: *p* ≥ 0.05; *: *p* < 0.05, **: *p* < 0.01).

**Figure 8 antioxidants-13-00525-f008:**
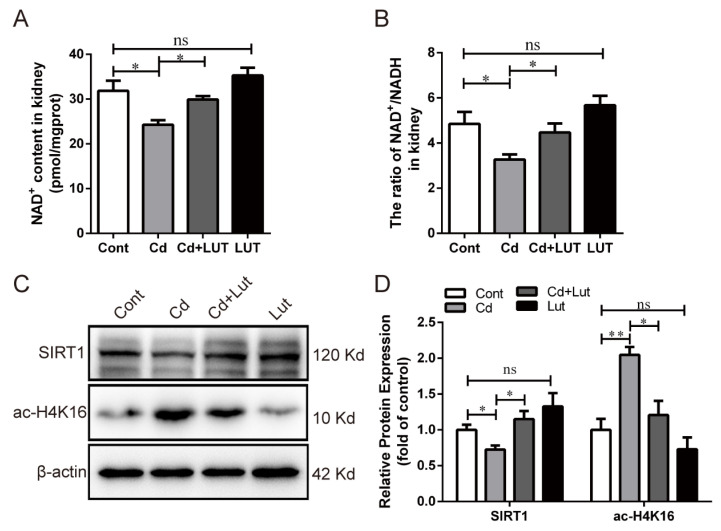
Effect of Lut and/or Cd on SIRT1 activity in chicken kidneys. (**A**) NAD^+^ content and (**B**) NAD^+^/NADH ratio in kidney tissues. (**C**,**D**) Western blotting images and quantitative analysis of SIRT1 and ac-H4K16. Each experiment was duplicated at least three times. (ns: *p* ≥ 0.05; *: *p* < 0.05, **: *p* < 0.01).

**Figure 9 antioxidants-13-00525-f009:**
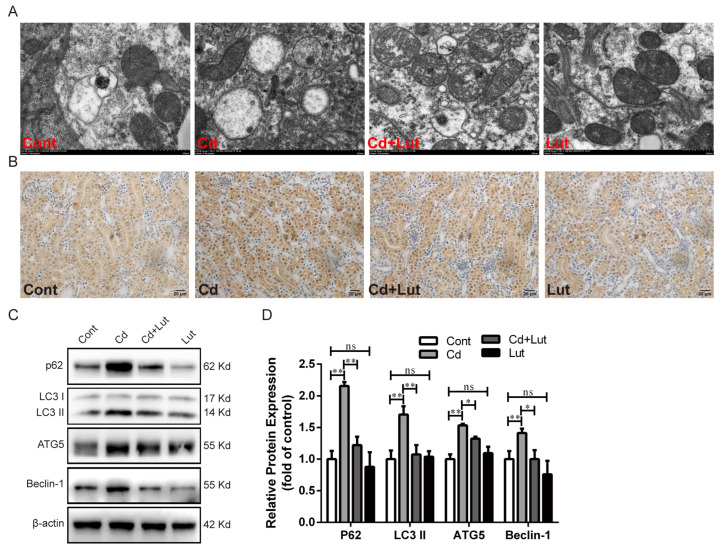
Effect of Lut and/or Cd on autophagy. (**A**) Observation of the number of autophagosomes or autolysosomes using TEM; scale bar: 500 nm. (**B**) LC3 protein expression was detected by the IHC; scale bar: 20 μm. (**C**,**D**) Western blotting and quantitative analysis of autophagy-related proteins p62, LC3 II, ATG5 and Beclin-1. Each experiment was duplicated at least three times. (ns: *p* ≥ 0.05; *: *p* < 0.05, **: *p* < 0.01).

## Data Availability

The datasets used and/or analyzed during the current study are available from the corresponding author on reasonable request.

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
