# Peer review of "Luteolin Alleviates Cadmium-Induced Kidney Injury by Inhibiting Oxidative DNA Damage and Repairing Autophagic Flux Blockade in Chickens"

_antioxidants, 2024, doi:10.3390/antiox13050525_

Round 1
Reviewer 1 Report
The manuscript submitted for review raises an important issue related to the threat of cadmium compounds, which can accumulate in the food chain, thus posing a threat to humans as well.
My detailed comments are as follows:
line 88 - Lut should be defined when used for the first time in the text
line 138 - "protective effect of Cd" - is that ok? Or "protective effect of Lut on Cd-induced kidney injury"?
Methods - in general, the scheme of the experiment is unclear.
How long did the experiment last?
Cd group - Was Cd administered once, on day 8 only?
Cd + Lut group - first Lut for 7 days, and then Cd for 1 month, is it right?
Lut group - For how many days was Lut adminsitered?
What was the form of Cd and Lut administered to chcickens?
point 2.3. - RTECs of which chicken? These from four groups presented above?
point 2.5 - When and how was blood collected?
Results:
Fig. 1 - O, 2.5 and 5 microM Cd not mentioned in methodology
Fig 2. - NAC not mentioned in methodology
Fig. 3 0 PJ36 not mentioned in methodology
Fig 4. RSV not mentioned in methodology
Reviewer 2 Report
The manuscript titled " Luteolin Alleviates Cadmium-Induced Kidney Injury by Inhibiting Oxidative DNA Damage and Repairing Autophagic Flux Blockade in Chickens " is well-written and presents an interesting topic. However, the manuscript requires modifications and additional information in the Methods and Materials section before it can be published. In this sense, same figures are not clear without additional information about the methods and materials used. The authors should revise the M&M section of manuscript to include this information.
For example:
Figure 1: Indicate on M&M how many animals are there in each group? and how? And for how long has 0, 1.25, 2.5 and 5 μM Cd been incorporated? Has it been given in water or feed? etc.
Figure 2. Where has the incorporation of NAC been explained? Indicate on M&M how many animals are in each group? and how? dose? and since when has NAC been given.
Figure 3. The same comment to Figure 1 and 3 for incorporation of PJ34.
Figure 4. The same comment to Figure 1 and 3 for incorporation of RSV.
Please, see previous comments.
Round 2
Reviewer 1 Report
Thank you very much for explanations and changes made in the text.
I am satisfied with the response and changes made.
Reviewer 2 Report
The authors present an interesting work with application in broiler production.The authors present an interesting work with application in broiler production.The authors present an interesting work with application in broiler production.
The authors have modified the manuscript on my recommendations.